# A Worldwide Bibliometric Analysis of Publications on Artificial Intelligence and Ethics in the Past Seven Decades

Chien-Wei Chuang [1,2,†] , Ariana Chang [3,†] , Mingchih Chen [1,2] , Maria John P. Selvamani [4,5] and Ben-Chang Shia [1,2,*]

1 Graduate Institute of Business Administration, Fu Jen Catholic University, New Taipei City 242062, Taiwan
2 Artificial Intelligence Development Center, Fu Jen Catholic University, New Taipei City 242062, Taiwan
3 Interdisciplinary Studies Program, Fu Jen Catholic University, New Taipei City 242062, Taiwan
4 School of Medicine, Fu Jen Catholic University, New Taipei City 242062, Taiwan
5 Fu Jen Academia Catholica, Fu Jen Catholic University, New Taipei City 242062, Taiwan
* Correspondence: 025674@mail.fju.edu.tw
† These authors contributed equally to this work.

**Abstract:** Issues related to artificial intelligence (AI) and ethics have gained much traction worldwide. The impact of AI on society has been extensively discussed. This study presents a bibliometric analysis of research results, citation relationships among researchers, and highly referenced journals on AI and ethics on a global scale. Papers published on AI and ethics were recovered from the Microsoft Academic Graph Collection data set, and the subject terms included "artificial intelligence" and "ethics." With 66 nations' researchers contributing to AI and ethics research, 1585 papers on AI and ethics were recovered, up to 5 July 2021. North America, Western Europe, and East Asia were the regions with the highest productivity. The top ten nations produced about 94.37% of the wide variety of papers. The United States accounted for 47.59% (286 articles) of all papers. Switzerland had the highest research production with a million-person ratio (1.39) when adjusted for populace size. It was followed by the Netherlands (1.26) and the United Kingdom (1.19). The most productive authors were found to be Khatib, O. (n = 10), Verner, I. (n = 9), Bekey, G. A. (n = 7), Gennert, M. A. (n = 7), and Chatila, R., (n = 7). Current research shows that research on artificial intelligence and ethics has evolved dramatically over the past 70 years. Moreover, the United States is more involved with AI and ethics research than developing or emerging countries.

**Keywords:** AI; ethics; bibliometric analysis; citation analysis; worldwide trend

## 1. Introduction

Artificial Intelligence (AI) has vastly disrupted people's daily lives and has had a profound effect on the way we live and work. Many "human" tasks can now be successfully performed by AI, and all sectors of the economy are being transformed by AI [1]. The question of whether AI is poised to disrupt or advance industries is of great debate. In recent years, there is escalating interest in the debate regarding AI's privacy and ethical issues [2]. It is imperative that we gain a comprehensive understanding of the AI and ethics landscape to determine the underlying mechanisms of the issues related to AI and ethics research. From a global vantage point, how countries contribute to the trajectories of AI and ethics research is of significance.

Studies have revealed an increasing willingness to utilize digital technology and Big Data in all industries [3]. Numerous industries in different sectors are expanding their investments in data-driven decision making and business analytics solutions to improve their performance and operations [4,5]. AI can apply human problem-solving behavior and skills to address complex real-world problems for better performance [6]. When leveraged critically, the development of AI can advance societal well-being and prevent risk [7].

Despite the advantages of AI applications, ethical concerns are still prevalent. Issues in regard to how we analyze, interpret, share, and replicate the data provided are frequently raised. The basis of this expanding attention on AI and ethics includes how it may affect human workers as technologies can increasingly execute jobs that were previously designated for humans, replacing a wide array of jobs [8–10]. Academia, governmental bodies, and private institutions have gained much traction in putting forward ethical principles, guidelines, statements, and various documents to provide direction on AI and ethics [11–13] due to malicious applications and abuses of AI. Therefore, ethical concerns regarding AI applications should not be dismissed.

AI has been identified as an emergent topic for empirical research [14]. Increasing concern regarding the impact of AI has prompted the emergence of the field of AI and ethics [15]. To the best of our knowledge, there has been scant research conducted on the discourse of AI and ethics research alone. In particular, the various strands of AI and ethics research have not been examined. This study sheds light on the global trends of AI and ethics research by utilizing bibliometric analyses. It is quintessential that we have an overall understanding on how different strands of research connect. To better understand the trend of publishing on this topic, we used the Microsoft Academic Graph database [16,17] to conduct analyses by the statistical method of the literature related to AI and ethics. By so doing, this study contributes to providing scholars new avenues for research on AI and ethics.

## 2. Methods

### 2.1. Literature Search

One of the ways to evaluate the academic publication of different countries is to present the total number of papers, the countries with the highest paper productivity, journals [18–21], and highly cited papers in tabular form, which can be utilized to investigate the worldwide trends of paper publications [22]. We chose papers related to AI and ethics recorded in the MAG in this bibliometric study. The literature search was not restricted to Science Citation Index Expanded and Social Science Citation Index Expanded. The research field included (Artificial intelligence) AND (ethics) and was refined to papers published from 1952 to 2021, without language limitations.

### 2.2. Data Analysis

To illustrate each country's research contribution and worldwide influence, we examined each country's publication production through descriptive statistics values such as the publication's sum of quantities, the sum of paper citations, the average number of paper citations, and the impact factor (IFs) [23]. All values correspond to the data included in the Microsoft Academic Graph, calculated as of 5 July 2021, including the sum of papers published, the number of papers cited, and the average number of papers cited. The quantity of citations in an article is often used to evaluate the influence of the academic study. The 2020 Journal Citation Reports of Clarivate Analytics were used to determine each journal's impact factor.

To extend the comparison between countries, we retrieved population [24], gross domestic product (GDP) [25], and Sustainable Development Report Score data from the International Monetary Fund (IMF), the United Nations (UN), and the Sustainable Development Report. To achieve a more sustainable future, the UN 2030 Agenda for Sustainable Development was adopted by all member states of the UN in 2015 [26]. The Sustainable Development Report measures each country's progress towards achieving the sustainable development goals (SDGs).

We measured countries' productiveness by using the following formula:

$$\frac{\text{publication numbers}}{\text{million populace}} \tag{1}$$

$$\frac{\text{publication numbers}}{\text{GDPs}} \qquad (2)$$

$$\frac{\text{publication numbers}}{\text{SDG Score}} \qquad (3)$$

In discussing the relationship between researchers, we use R's supplementary package "visNetwork" [27] and Microsoft PowerBI to visualize the relationship between researchers and establish a social network according to the amount of cooperative publishing, forming a huge academic network map. Figure 1 below represents the bibliometric process implemented in this study.

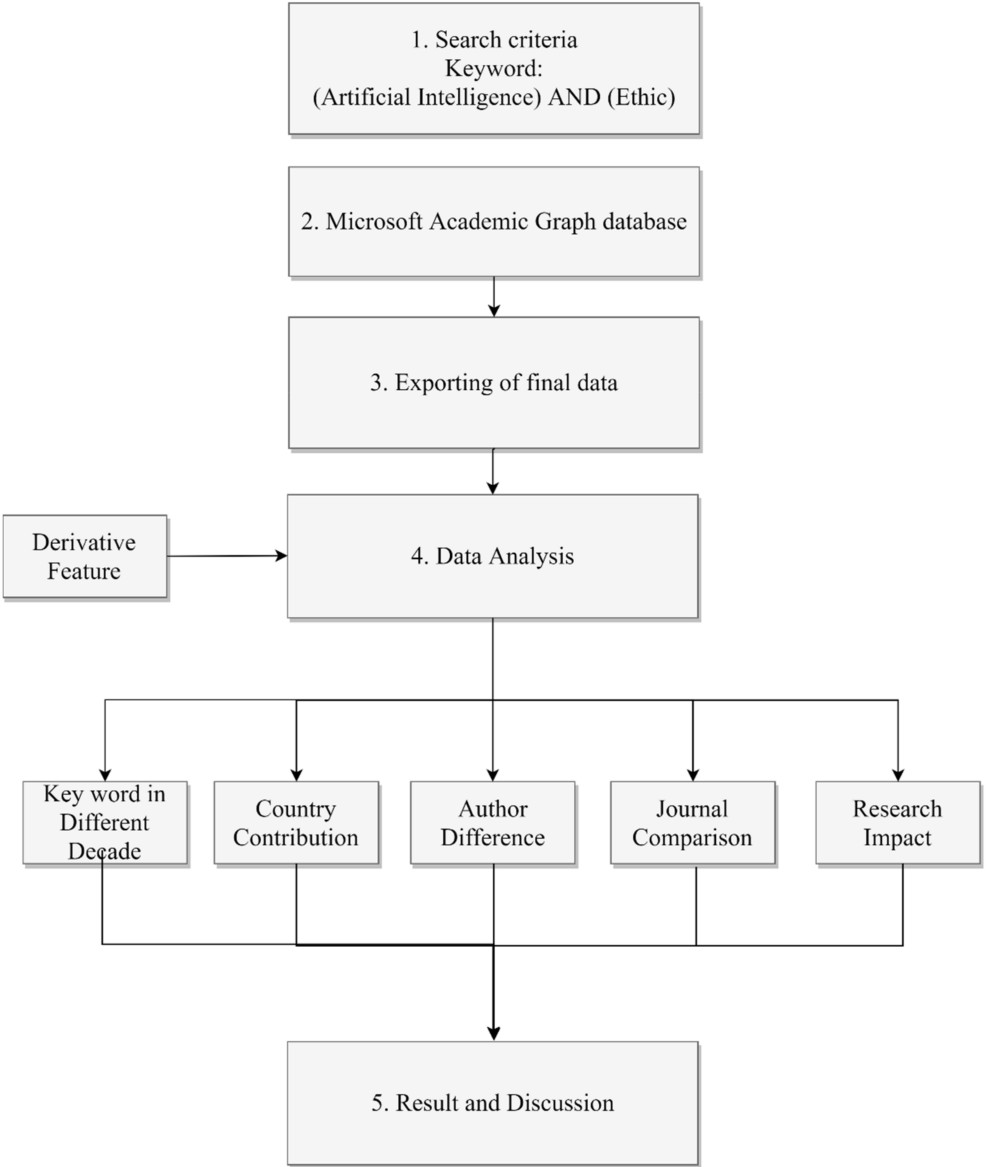

**Figure 1.** Research Framework flow chart.

## 3. Results

### 3.1. Worldwide Trends of Academic Publication

In the last seven decades, about 1585 published papers were contained within the MAG index. Sixty-six countries worldwide have contributed to AI and ethics research (Figure 2). North America is the place with the most papers published, followed by Western Europe and East Asia. Only one country published more than 100 articles, and 11 countries published more than ten articles. A total of 166 papers related to artificial intelligence

and ethics were published before 1990, 147 papers were published between 1990 and 1999, 453 papers were published in the first decade of the 21st century, and the number of publications increased between 2010 to 2019, to 720 papers (Figure 3).

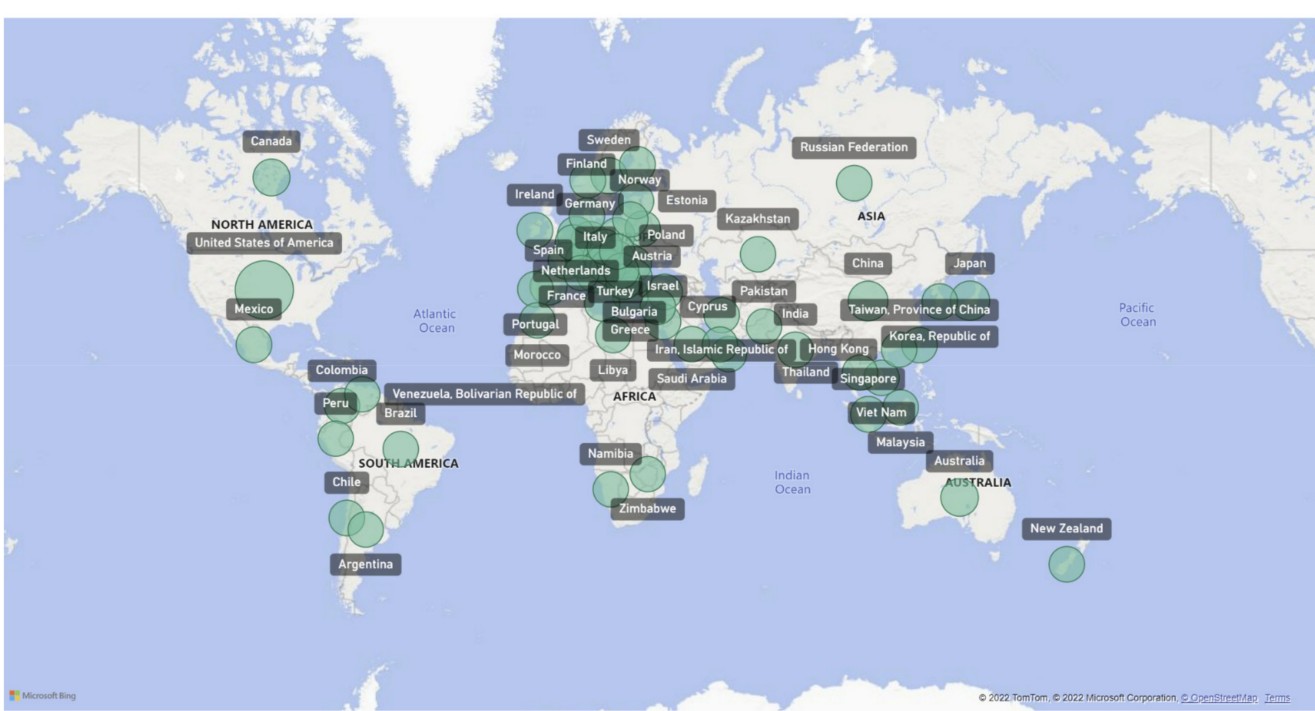

**Figure 2.** World map of paper production by countries and regions.

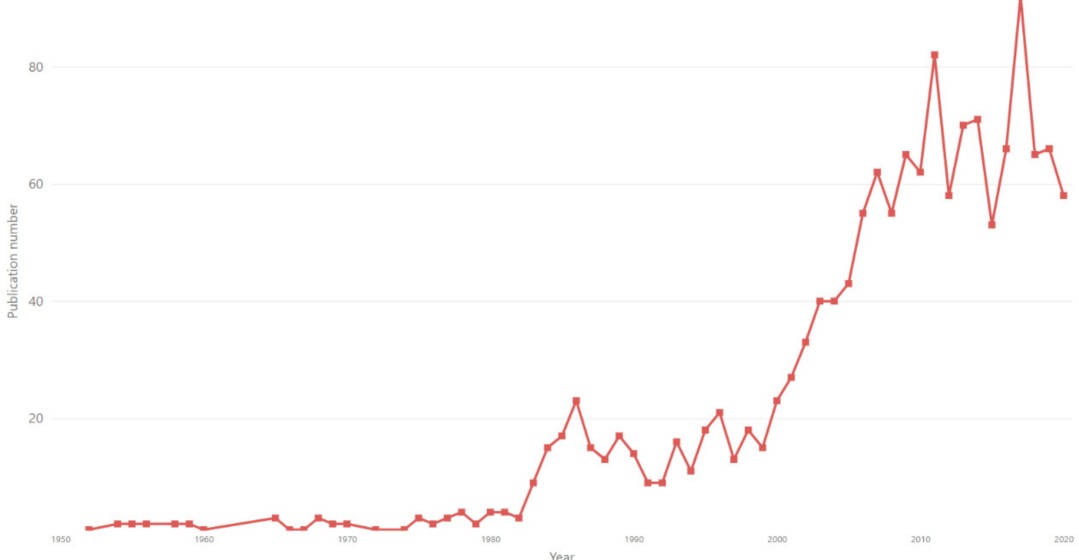

**Figure 3.** Worldwide publication on AI and ethics from 1950–2020.

### 3.2. AI and Ethics Research Publication Count by Country

We have compiled a table to present the current publication status of each country and rank the top 20 countries by publishing quantity. Articles only with registered countries in the MAG database will be classified. According to the calculation under this definition, the United States has the largest number of publications. A total of 286 papers have been published, accounting for 47.59%, followed by The United Kingdom which has a total of 80 papers, accounting for 13.31%. Third is China, which has published 56 papers, accounting for 9.32%, and fourth is Japan, which has published 51 papers, accounting for

8.49%. According to the grouping of nominal gross national income per capita defined by the World Bank, except for Brazil, China, India, Mexico, and Russia, which have upper-middle-income economies, all other countries and regions have high-income economies. According to the ranking in Table 1, the top 20 countries accounted for 95.9% of the world's research publications.

**Table 1.** Publication and Citation Analysis with Top 20 Productive Countries.

| Nations | Paper Publish Number | % | Citing Articles | Average Citations | Population (in Millions) | Number of Articles Per Million Inhabitants | GDP (US $1000B) | Number of Articles Per GDP (US $1000B) | SDG Score | Articles Per SDG Score |
|---|---|---|---|---|---|---|---|---|---|---|
| United States of America | 286 | 47.59% | 11,599 | 40.556 | 329.48 | 0.87 | 209.37 | 1.366 | 76.01 | 3.763 |
| United Kingdom of Great Britain and Northern Ireland | 80 | 13.31% | 1608 | 20.100 | 67.2 | 1.19 | 27.08 | 2.954 | 79.97 | 1 |
| China | 56 | 9.32% | 47 | 0.839 | 1443.5 | 0.04 | 147.23 | 0.38 | 72.06 | 0.777 |
| Japan | 51 | 8.49% | 320 | 6.275 | 125.67 | 0.41 | 50.65 | 1.007 | 79.85 | 0.639 |
| Germany | 37 | 6.16% | 379 | 10.243 | 83.17 | 0.44 | 38.06 | 0.972 | 82.48 | 0.449 |
| Italy | 31 | 5.16% | 794 | 25.613 | 59.64 | 0.52 | 18.86 | 1.644 | 78.76 | 0.394 |
| Australia | 29 | 4.83% | 846 | 29.172 | 25.68 | 1.13 | 13.31 | 2.179 | 75.58 | 0.384 |
| Spain | 23 | 3.83% | 312 | 13.565 | 47.33 | 0.49 | 12.81 | 1.795 | 79.46 | 0.289 |
| Netherlands | 22 | 3.66% | 336 | 15.273 | 17.41 | 1.26 | 9.12 | 2.412 | 81.56 | 0.27 |
| Canada | 20 | 3.33% | 84 | 4.200 | 38.01 | 0.53 | 16.43 | 1.217 | 79.16 | 0.253 |
| France | 16 | 2.66% | 110 | 6.875 | 67.29 | 0.24 | 26.03 | 0.615 | 81.67 | 0.196 |
| Switzerland | 12 | 2.00% | 338 | 28.167 | 8.61 | 1.39 | 7.48 | 1.604 | 80.1 | 0.15 |
| Poland | 7 | 1.16% | 8 | 1.143 | 37.96 | 0.18 | 5.94 | 1.178 | 80.22 | 0.087 |
| India | 7 | 1.16% | 5 | 0.714 | 1347.12 | 0.01 | 26.23 | 0.267 | 60.07 | 0.117 |
| Mexico | 6 | 1.00% | 6 | 1.000 | 126.01 | 0.05 | 10.76 | 0.558 | 69.13 | 0.087 |
| Russian Federation | 5 | 0.83% | 9 | 1.800 | 146.2 | 0.03 | 14.84 | 0.337 | 73.75 | 0.068 |
| Turkey | 4 | 0.67% | 5 | 1.250 | 83.61 | 0.05 | 7.2 | 0.556 | 70.38 | 0.057 |
| Brazil | 4 | 0.67% | 5 | 1.250 | 211.82 | 0.02 | 14.45 | 0.277 | 71.34 | 0.056 |
| Korea, Republic of | 3 | 0.50% | 10 | 3.333 | 51.78 | 0.06 | 16.31 | 0.184 | 78.59 | 0.038 |
| Saudi Arabia | 2 | 0.33% | 4 | 2.000 | 35 | 0.06 | 7 | 0.286 | 66.3 | 0.03 |

Abbreviations: GDP (Gross domestic product).

A total of 48,735 citations were made in these 1585 papers. The average number of citations per paper was 30.75. The United States, Australia, and Switzerland were the three countries with the highest number of citations, with average citations of 40.75, 29.17, and 28.17, respectively. Additional comparisons were made in proportion to the current situation in different countries. First, the number of publications was adjusted for population data. Dividing the number of publications by the population showed that Switzerland had the highest publications per million inhabitants, at 1.39, followed by the Netherlands at 1.26 and the UK at 1.19. Second, the number of paper publications was adjusted for GDP data by dividing the number of publications by the GDP (in 1000 billion). The highest ratio was the UK at 2.954, followed by the Netherlands at 2.412 and Australia at 2.179. Third, the number of publications was adjusted by SDG Scores data by dividing the number of papers published by the SDG score. The highest ratio of papers divided by SDG score was in the United States at 3.763, followed by the UK at 1 and China at 0.777.

### 3.3. Journal Publishing Comparison

In the past 70 years, 1072 papers have been published in 732 journals, of which 596 are journals included in SCI and SSCI. As Table 2 all journals, only one journals published more than 50 papers, and six journals published more than ten papers. Among all journals, 476 journals published only one paper.

**Table 2.** Top 10 SCI & SSCI Journals Publishing AI and Ethics Articles.

| Journal Names | Articles | % | Total Citation | Mean Citation Per Article | Journal Impact Factor | Impact Factor without Journal Self Cites | 5-Year Impact Factor |
|---|---|---|---|---|---|---|---|
| IEEE Robotics and Automation Magazine | 63 | 10.57% | 632 | 10.0317 | 3.591 | 3.466 | 4.615 |
| The International Journal of Robotics Research | 33 | 5.54% | 18 | 0.5455 | 4.703 | 4.479 | 6.397 |
| AI Magazine | 21 | 3.52% | 338 | 16.0952 | 1.627 | 1.56 | 1.742 |
| Industrial Robot: An International Journal | 17 | 2.85% | 335 | 19.7059 | 1.123 | 0.911 | 1.287 |
| Advanced Robotics | 12 | 2.01% | 9 | 0.75 | 1.247 | 1.184 | 1.215 |
| Communications of The ACM | 12 | 2.01% | 474 | 39.5 | 6.988 | 6.844 | 6.064 |
| International Journal of Social Robotics | 9 | 1.51% | 279 | 31 | 2.516 | 2.108 | 3.168 |
| Connection Science | 8 | 1.34% | 39 | 4.875 | 1.042 | 1 | 1.191 |
| Journal of the Royal Society of Medicine | 8 | 1.34% | 159 | 19.875 | 5.238 | 3.905 | 3.405 |
| Kybernetes | 7 | 1.17% | 20 | 2.8571 | 1.754 | 1.498 | 1.47 |

Abbreviations: AI, artificial intelligence; IF, impact factor.

Among the journals with Impact factor, IEEE Robotics & Automation Magazine has published 63 papers, accounting for 10.57%. The second is The International Journal of Robotics Research, which has published 33 papers, accounting for 5.54%. The third is AI Magazine, which published 21 papers, accounted for about 3.52%.

### 3.4. Role of AI and Ethics across Disciplines

The role of AI and ethics could have a negative influence on policy since it frequently dismisses ethical concerns related to prejudice, information asymmetry, and the ramifications of digital interactions in the twenty-first century.

While AI and ethics has become an important topic in AI research, it has spanned a variety of disciplines. In Table 3 we can see that since the 1980s, the disciplines of engineering and computer science have remained the top disciplines for AI and ethics research. It is worthy to note that the exploration of AI and ethics in sociology research has increased incrementally. Sociology became the second most widely covered discipline in AI and ethics research from 2020 to 2021. As the inequalities embedded in our society have become more prevalent, there is a need to examine AI and ethics research from the vantage points of sociology. In our social interface, our research disciplines have become "technologically" fused.

The keywords that have topped the keyword research include robotics, robot, and curriculum. From the top keyword list, there is a clear link to the discipline of engineering. The COVID-19 pandemic has also spurred the keyword Coronavirus disease 2019. Results are presented in Table 4.

**Table 3.** AI and Ethics keyword analysis with disciplines.

| Year | 1950~1954 | 1955~1959 | 1960~1964 | 1965~1969 | 1970~1974 | 1975~1979 | 1980~1984 | 1985~1989 | 1990~1994 | 1995~1999 | 2000~2004 | 2005~2009 | 2010~2014 | 2015~2019 | 2020~2021 |
|---|---|---|---|---|---|---|---|---|---|---|---|---|---|---|---|
| Rank1 | Computer science | Computer science | Medicine | Engineering | Computer science | Computer science | Engineering | Engineering | Engineering | Engineering | Engineering | Engineering | Engineering | Engineering | Engineering |
| Rank2 | Psychology | Engineering | – | Computer science | Engineering | Psychology | Computer science | Computer science | Computer science | Computer science | Computer science | Computer science | Computer science | Computer science | Sociology |
| Rank3 | – | Psychology | – | – | – | Engineering | Medicine | Medicine | Psychology | Sociology | Psychology | Psychology | Psychology | Sociology | Computer science |
| Rank4 | – | – | – | – | – | Medicine | Sociology | Psychology | Sociology | Medicine | Sociology | Sociology | Sociology | Political science | Psychology |
| Rank5 | – | – | – | – | – | Political science | – | Political science | Medicine | Economics | Medicine | Medicine | Medicine | Psychology | Political science |
| Rank6 | – | – | – | – | – | Sociology | – | Sociology | Business | Psychology | Mathematics | Political science | Political science | Medicine | Medicine |
| Rank7 | – | – | – | – | – | – | – | – | Mathematics | – | – | Business | Philosophy | Business | – |

**Table 4.** AI and Ethics keyword analysis without disciplines.

| Year | 1950~1954 | 1955~1959 | 1960~1964 | 1965~1969 | 1970~1974 | 1975~1979 | 1980~1984 | 1985~1989 | 1990~1994 | 1995~1999 | 2000~2004 | 2005~2009 | 2010~2014 | 2015~2019 | 2020~2021 |
|---|---|---|---|---|---|---|---|---|---|---|---|---|---|---|---|
| Rank1 | Alternative medicine | Computer technology | MEDLINE | Curriculum | Public opinion | Alternative medicine | Robotics | Robotics | Robotics | Robotics | Robotics | Robotics | Robotics | Robotics | Robotics |
| Rank2 | – | Technological revolution | – | Scientific discovery | Terminology | Emerging technologies | Robot | Robot | Curriculum | Robot | Robot | Robot | Robot | Robot | Robot |
| Rank3 | – | – | – | – | – | Human rights | Automation | Alternative medicine | Expert system | Curriculum | Curriculum | Curriculum | Curriculum | Roboethics | Deep learning |
| Rank4 | – | – | – | – | – | – | Computer technology | Automation | Documentation | Pedagogy | Artificial life | Creativity | Roboethics | Deep learning | Social robot |
| Rank5 | – | – | – | – | – | – | Creativity | Acquired immunodeficiency syndrome (AIDS) | Robot | Alternative medicine | Creativity | Ethical issues | Human–robot interaction | Curriculum | Curriculum |
| Rank6 | – | – | – | – | – | – | Curriculum | Cognitive science | Artificial life | Evidence-based medicine | Natural language processing | Health care | Social robot | Social robot | Coronavirus disease 2019 (COVID-19) |
| Rank7 | – | – | – | – | – | – | Engineering management | Curriculum | Automation | Artificial life | Consciousness | Human–robot interaction | Multidisciplinary approach | Automation | Health care |
| Rank8 | – | – | – | – | – | – | Expert system | Expert system | Computer technology | Automation | Emerging technologies | Informatics | Automation | Autonomy | Machine ethics |
| Rank9 | – | – | – | – | – | – | Health informatics | Living systems | Creativity | Engineering management | Information science | Machine ethics | Pedagogy | Human–robot interaction | Multidisciplinary approach |
| Rank10 | – | – | – | – | – | – | Informatics | Medical ethics | Engineering management | Health technology | Information system | Multidisciplinary approach | Creativity | Health care | Sustainability |
| Rank11 | – | – | – | – | – | – | Information science | Artificial life | Health care | Information system | Terminology | Pedagogy | Health care | Creativity | Emerging technologies |
| Rank12 | – | – | – | – | – | – | Natural language processing | Engineering management | Health informatics | Autonomy | Alternative medicine | Automation | Knowledge management | Machine ethics | Human–robot interaction |
| Rank13 | – | – | – | – | – | – | Pedagogy | Environmental ethics | Information system | Deep learning | China | Civilization | Machine ethics | Multidisciplinary approach | Ai ethics |
| Rank14 | – | – | – | – | – | – | Terminology | Health care | Natural language processing | Health care | Computer technology | Health informatics | Terminology | Knowledge management | Autonomy |
| Rank15 | – | – | – | – | – | – | Medical imaging | Pedagogy | Environmental ethics | Expert system | Multidisciplinary approach | Knowledge management | Autonomy | Sustainability | Medical ethics |

### 3.5. Most Cited Papers and Publish Distribution

Table 5 shows the top ten cited papers. According to the MAG record, the most cited paper was published in Annals of Surgery in 2004, which has a very high citation count of 1267. Table 6 presents the situation and distribution of papers' citations. One way to measure the impact of each paper is through citation analysis, which calculates the number of citations as the paper's impact. Of the 1585 AI and ethics research publications, 43 papers were cited 50 times or more which is about 2.71%, and 60 papers were cited 100 times or more, about 3.79%. A total of 51.17% of papers were cited zero times.

**Table 5.** Top 10 Cited Papers on AI and ETHICS.

| Title | AuthorName | Country/Region | VenueName | Year | Citation Count |
|---|---|---|---|---|---|
| Robotic surgery: a current perspective | Anthony R. Lanfranco | United States of America | Annals of Surgery | 2004 | 1267 |
| All models are wrong: reflections on becoming a systems scientist | John D. Sterman | United States of America | System Dynamics Review | 2002 | 1266 |
| Engineering Education and the Development of Expertise | Thomas A. Litzinger | – | Journal of Engineering Education | 2011 | 566 |
| Biomimetics—Using nature to inspire human innovation | Yoseph Bar-Cohen | United States of America | Bioinspiration & Biomimetics | 2006 | 453 |
| Going digital: a look at assumptions underlying digital libraries | David N. L. Levy | United States of America | Communications of the ACM | 1995 | 410 |
| Prolegomena to any future artificial moral agent | Colin Allen | United States of America | Journal of Experimental and Theoretical Artificial Intelligence | 2000 | 359 |
| Surgical robotics: the early chronicles: a personal historical perspective | Richard M. Satava | United States of America | Surgical Laparoscopy Endoscopy and Percutaneous Techniques | 2002 | 335 |
| Exoskeletons and robotic prosthetics: a review of recent developments | Robert Bogue | – | Industrial Robot: An International Journal | 2009 | 320 |
| Complexity Theory in Organization Science: Seizing the Promise or Becoming a Fad? | Bill McKelvey | United States of America | Emergence | 1999 | 311 |
| Control: A perspective | Karl Johan Åström | – | Automatica | 2014 | 270 |

Abbreviation: AI, artificial intelligence; ACM, Association for Computing Machinery.

**Table 6.** Citation Distribution.

| No. | Number of Citations | Number of Papers | % |
|---|---|---|---|
| 1 | 0 | 811 | 51.17% |
| 2 | 1–10 | 534 | 33.69% |
| 3 | 11–50 | 137 | 8.64% |
| 4 | 51–100 | 43 | 2.71% |
| 5 | >100 | 60 | 3.79% |

### 3.6. Author Collaboration Relationship Analysis

We analyzed and visualized each author's publication and collaboration relationship using the R package "visNetwork." Only authors who had published a minimum of three papers are displayed on the network visualization map (Figure 4). The circle size is set based on the number of papers published by each author as the primary reference, and the line between the two authors represents the line of cooperation between them. Different colors represent collaboration clusters among different authors, while the same color indicates more frequent and closer collaboration. In the network visualization map, you can see that the authors with the highest number of publications are Oussama Khatib (n = 10), Igor M. Verner (n = 9), George A. Bekey (n = 7), Michael A. Gennert (n = 7), and Raja Chatila (n = 7).

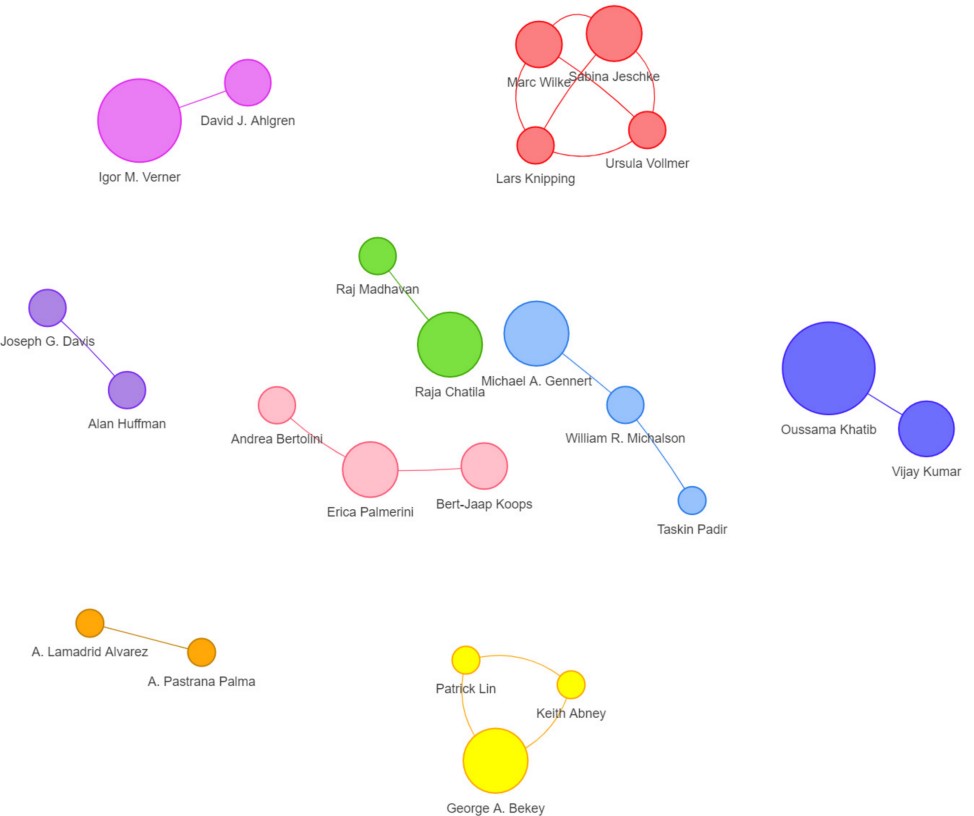

**Figure 4.** Authorship collaboration network visualization map.

### 3.7. Institutions Collaboration Relationship Analysis

We used Microsoft Power BI's Network Navigator Chart to visualize the network, which includes research that has been submitted by institutions. The node size is set according to the weighted number of papers published by each research institution as the primary reference, and the line between two research institutions represents the collaboration between them. The different colors represent collaborative clusters between different research institutions, while the same color represents more frequent and tighter collaborations. In the grouping process, research units that have published at least six papers are included in the calculation. After a preliminary screening of 554 research institutions, the remaining 306 research institutions are those that have published more than six papers. The samples were grouped into 57 groups through cluster analysis, and the final appearance is shown in Figure 5. Group 48 is the largest publishing group among all groups, with Johns Hopkins University, Technische Universität München, Heidelberg University, Imperial College London, and Harvard University as the core, and the color is PowderBlue. The second largest group is Group 23; this group is mainly composed of the National University of Distance Education, the Charles III University of Madrid, the University of Zaragoza, and the University of Applied Sciences Stuttgart; the color is Lavender.

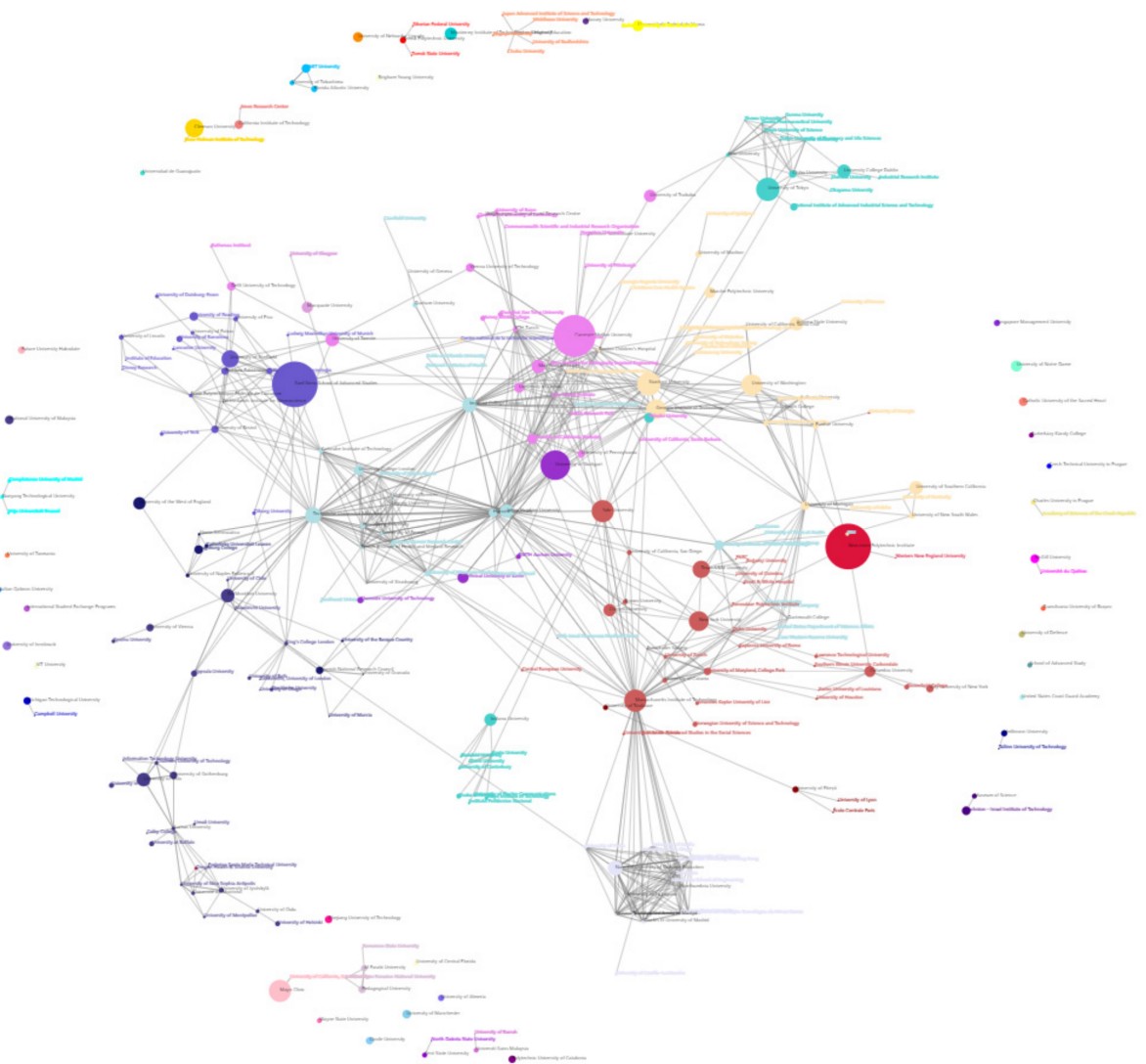

**Figure 5.** Network visualization map of collaborations between institutions.

## 4. Discussion

Bibliometrics analysis can analyze articles, books, and other publications which are frequently used in library and information science. Thus, it comprises a collection of approaches to assess scientific documents in areas such as science and technology studies [21]. The outcomes from the bibliometric analysis provide us with data regarding research action patterns over the long term and measure the quantified result of individual researchers, affiliations, journals, institutions, and nations. Other areas of analysis incorporate the fields of interest, the country's published contributions, the top journals, author collaboration relationships, paper citation analysis, publication growth situations, and most-cited papers in the specified research field.

Bibliometric research uses quantitative methods to conduct statistical analysis in order to explore the current state of research, measure the degree of impact of research from different aspects, and demonstrate the overall global patterns of the field. Utilizing bibliometric analysis enables us to explore the subtleties of a particular field's evolutionary history while drawing out the emerging areas of study [22]. Such a research method can guide us to grasp how AI and ethics research has evolved. Most importantly, it can help us determine potential avenues of future research.

This study reviewed the current status and trends of artificial intelligence and ethics research. In total, scholars from 66 countries contributed to the publishing of papers related

to artificial intelligence and ethics. The United States published the most papers among all nations and in all countries. American researchers published 80% of the top ten most cited papers. In terms of average citations per article, the United States is also the best performer of all countries, with an average of 40.75 citations per paper. After adjusting for population size, Switzerland had the highest rate of any country with 1.39 papers published per million inhabitants. Adjusted using the GDP, the UK had the highest rate, at 0.0295 papers. AI and ethics are topics that have been paid more and more attention in recent years and will continue to receive attention in the future.

According to the Sustainable Development Report, in this study, we found that the United States had a much higher percentage of articles on sustainability than other countries, indicating that the United States pays more attention to the development of AI and ethics than to that of sustainable development. The United Kingdom has a ratio of 1, which means that the degree of concern is approximately the same. However, China, Japan, Germany, and other countries are more concerned about sustainable development than AI and ethics.

In this study, we also found that most countries that study artificial intelligence and ethics are medium-high and highly developed countries. The United States has the most published papers and citations, which means that American scholars have a certain degree of influence in AI and ethics. It is worth noting that China's performance ranks in the top three in terms of publication volume, with 56 papers published, but it does not perform well in the average number of citations, at only 0.84 citations. Another country worth mentioning is Switzerland. Switzerland has only 12 publications, but the average number of citations is 28.17 times, and each paper is cited relatively often.

Among the cited articles in the past 70 years, the results are mostly related to biomedical, scientific, and technological engineering fields. The top ten most cited papers are within the fields of medical biology and scientific and technical engineering. The medical and biological journals are Annals of Surgery, Bioinspiration & Biomimetics, Surgical Laparoscopy Endoscopy and Percutaneous Techniques, and Emergence. The journals related to science and technology engineering are System Dynamics Review, Journal of Engineering Education, Communications of the ACM, Journal of Experimental and Theoretical Artificial Intelligence, Industrial Robot: An International Journal, and Automatica. The research and development of AI and ethics are also generally mostly applied within the fields of medical, scientific, and technological engineering, so these articles have received higher citations.

How the impact of AI and ethical research will benefit different countries, cultures, populations, and races is currently uncertain. In future research, we hope to explore beyond the context of countries, and consider people and race, further increasing the complexity of our research. In this way, the literature on AI and ethics can be made more abundant.

This study is the first to attempt to focus on the bibliometric analysis of artificial intelligence and ethics. This study still has limitations. First, in this study, we only used the MAG database to search for publications, and papers not included in the MAG were not counted. Second, there is a possibility of bias due to the number of citations recorded by MAG.

## 5. Conclusions

In today's turbulent times, the implications of understanding the realms of AI and ethics research are imperative. Tracing back to see how AI and ethics research has evolved over the years can benefit us in foreseeing future trends. Although AI research has gained significant traction since the 1990s, this study has illustrated the actual output of AI and ethics research across the world. The results of the study show that AI and ethics research spans across multiple disciplines. The results also indicate that the majority of AI and ethics research has been conducted in the discipline of engineering. Since the 1980s, it has remained the discipline with the highest coverage of AI and ethics. Specifically, this research demonstrates that engineering-related AI applications continue to be plagued by ethical issues.

With the most published research articles, the United States and the United Kingdom lead AI and ethics research, accounting for 61% of all publications. Australia, Switzerland, and Italy also have considerable influence. It is interesting to note that those more engaged in AI and ethics are from highly developed countries. Our findings contribute to enriching the discussion of AI and ethics by thoroughly examining various aspects of AI and ethics research to understand why it is focused in a particular discipline. This study contributes to the emerging agenda on the evolution of AI and ethics research.

**Author Contributions:** Conceptualization, B.-C.S. and M.J.P.S.; methodology, A.C. and C.-W.C.; software, C.-W.C.; validation, A.C., C.-W.C. and B.-C.S.; formal analysis, C.-W.C.; investigation, A.C. and C.-W.C.; resources, B.-C.S. and C.-W.C.; data curation, C.-W.C.; writing—original draft preparation, A.C. and C.-W.C.; writing—review and editing, A.C.; visualization, C.-W.C.; project administration, M.C.; funding acquisition, B.-C.S. and M.J.P.S. All authors have read and agreed to the published version of the manuscript.

**Funding:** This manuscript was partially funded by Grant number: 7100397 and Grant number: A0110152.

**Institutional Review Board Statement:** Not applicable.

**Informed Consent Statement:** Not applicable.

**Data Availability Statement:** Not applicable.

**Conflicts of Interest:** The authors declare no conflict of interest.

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
