# Peer review of "A Worldwide Bibliometric Analysis of Publications on Artificial Intelligence and Ethics in the Past Seven Decades"

_sustainability, doi:10.3390/su141811125_

Round 1

Reviewer 1 Report

Thank you for the opportunity to read the paper. 

Within the limits of the possibilities, the following mapping can be done:

  • Mapping of co-occurrence author keywords
  • Mapping of articles co-citations
  • Mapping of references co-citations
  • Mapping of journal co-citations
  • Mapping of institutions’ co-citations
  • Mapping of countries co-authorships
  • Mapping of countries bibliographic coupling

The bibliography can be improved. Possible sources can be:

Radu, V.; Radu, F.; Tabirca, A.I.; Saplacan, S.I.; Lile, R. Bibliometric Analysis of Fuzzy Logic Research in International Scientific Databases. Int. J. Comput. Commun. Control 2021, 16, 1–20.

Author Response

We are extremely grateful to the Reviewer 1 for the helpful and insightful comments and suggestions provided on the overall version of our manuscript. We have made every effort to take all of the constructive suggestions, and believe that due the changes that have now been made, the latest revision of the paper has been improved immensely.

The major changes made in the latest version of our paper are summarized as follows: (i) the contribution and novelty of this research are more defined; (ii) new mapping is generated; and (iii) theoretical links and sources are elaborated.

The specific point-by-point revisions made in response to the suggestions provided by Reviewer 1 are provided below:

  1. Within the limits of the possibilities, the following mapping can be done

Response:

A “Network visualization map of collaborations between institutions” has been conducted to visualize research that have been submitted by institutions. Please see Figure 5.

  1. The bibliography can be improved.

Response:

The following references have been added to explain the theoretical links.

Radu, V.; Radu, F.; Tabirca, A.I.; Saplacan, S.I.; Lile, R. Bibliometric Analysis of Fuzzy Logic Research in International Scientific Databases. Int. J. Comput. Commun. Control 2021, 16, 1–20.

Thank you very much for your time.

Reviewer 2 Report

The authors have revised the manuscript by adding new and relevant information. Overall, it appears to be a detailed bibliometric study of AI and ethics, which will be useful to the research community.

Author Response

1. The authors have revised the manuscript by adding new and relevant information. Overall, it appears to be a detailed bibliometric study of AI and ethics, which will be useful to the research community.

Response

1. Thank you reviewer for your valuable comments to make the article more comprehensive.

Reviewer 3 Report

·         The novelty of this paper is not clear. The difference between present work and previous Works should be highlighted.

·         The author needs to change the abstract and focus more on problem domain. Before the paper contributions, the author could precisely include the need of developing the proposed method.

·        

·         The author could better explain how “Related works” is actually related to the current study. It is not clear to the reader how the manuscript is similar to or differs from these related works.

·         Is there any limitation of the proposed work? If so, the author should include it at the end of conclusion part. This may help future researchers to overcome the limitations.

·          Some recent works should be added, such as: https://doi.org/10.1007/s10586-021-03472-

·         Authors must develop the framework/architecture of the proposed methods

·         Flowcharts are missed in the paper, also there is no algorithm. It is suggested to write an algorithm with necessary time complexity.

Author Response

We are extremely appreciative to the Reviewer 3 for the constructive criticisms of our paper. We have made every effort to edit the paper to reflect the majority of the reviewer’s recommendations. Due to the changes made, we believe that the latest revision has been improved greatly. Here is a point-by-point response to the comments and concerns.

The specific point-by-point revisions made in response to the suggestions provided by Reviewer 3 are provided below:

  1. The novelty of this paper is not clear. The difference between present work and previous works should be highlighted.

Response: The following references have been added to explain and enhance the theoretical links. Thus, these theoretical links shed highlights the differences between our work and previous works. To shed light on the novelty of this research, the conclusion has been adjusted as well.

Brundage, M., Avin, S., Clark, J., Toner, H., Eckersley, P., Garfinkel, B., … & Amodei, D. (2018). The malicious use of artificial intelligence: Forecasting, prevention, and mitigation. arXiv preprint arXiv:1802.07228.

Castañeda, K., Sánchez, O., Herrera, R. F., & Mejía, G. (2022). Highway Planning Trends: A Bibliometric Analysis. Sustainability, 14(9), 5544.

Demir, E., YaÅŸar, E., Özkoçak, V., & Yıldırım, E. (2020). The evolution of the field of legal medicine: A holistic investigation of global outputs with bibliometric analysis. Journal of Forensic and Legal Medicine69, 101885.

Donthu, N., Kumar, S., Mukherjee, D., Pandey, N., & Lim, W. M. (2021). How to conduct a bibliometric analysis: An overview and guidelines. Journal of Business Research133, 285-296.

Fiandrino, S., Scarpa, F., & Torelli, R. (2022). Fostering Social Impact Through Corporate Implementation of the SDGs: Transformative Mechanisms Towards Interconnectedness and Inclusiveness. Journal of Business Ethics, 1-15.

Mela, G., Martinoli, C., Poggi, E., & Derchi, L. (2003). Radiological research in Europe: a bibliometric study. European radiology13(4), 657-662.

Merigó, J. M., Pedrycz, W., Weber, R., & de la Sotta, C. (2018). Fifty years of Information Sciences: A bibliometric overview. Information Sciences432, 245-268.

Jobin, A., Ienca, M., & Vayena, E. (2019). The global landscape of AI ethics guidelines. Nature Machine Intelligence1(9), 389-399.

Ryan, M., & Stahl, B. C. (2020). Artificial intelligence ethics guidelines for developers and users: clarifying their content and normative implications. Journal of Information, Communication and Ethics in Society.

Stahl, B. C., Antoniou, J., Ryan, M., Macnish, K., & Jiya, T. (2022). Organisational responses to the ethical issues of artificial intelligence. AI & SOCIETY, 37(1), 23-37.

Wilson, H. J., & Daugherty, P. R. (2018). Collaborative intelligence: Humans and AI are joining forces. Harvard Business Review, 96(4), 114-123.

Zou, J., & Schiebinger, L. (2018). AI can be sexist and racist—it’s time to make it fair.

  1. The author needs to change the abstract and focus more on problem domain. Before the paper contributions, the author could precisely include the need of developing the proposed method.

Response: The research method is extended to explain why bibliometric analysis is necessary for this study. Please see the methodology section. This method has been employed to better grasp the comprehensive picture of AI and ethics research.

  1. The author could better explain how “Related works” is actually related to the current study. It is not clear to the reader how the manuscript is similar to or differs from these related works.

Response: Theoretical links have been added and such links are the related works that are relevant to this study.

  1. Is there any limitation of the proposed work? If so, the author should include it at the end of conclusion part. This may help future researchers to overcome the limitations.

Response: First, in this study, we only use the MAG database to search for publications, and papers not included in MAG will not be counted. Second, there is a possibility of bias due to the number of citations recorded by MAG. Not all works in every possible language are included in this study.

  1. Some recent works should be added.

Response: Theoretical links relevant to our work have been incorporated to enhance this study.

  1. Authors must develop the framework/architecture of the proposed methods

Response: New mappings such as the network visualization map between institutions have been incorporated. Please see Figure 5. Microsoft Power BI's Network Navigator Chart was used to visualize the network, which includes research that have been submitted by institutions. Collaborative clusters are exhibited by the different colors to represent the various research institutions.

  1. Flowcharts are missed in the paper, also there is no algorithm. It is suggested to write an algorithm with necessary time complexity.

Response: A conceptual framework has been added. Please refer to page 4. Thank you for your suggestion on writing an algorithm for this research. In this study, we directly searched for keywords in the data base, so we did not use search engine related algorithms.

Thank you very much for your time. Your comments have been invaluable.

This manuscript is a resubmission of an earlier submission. The following is a list of the peer review reports and author responses from that submission.

Round 1

Reviewer 1 Report

Thank you for the opportunity to read the paper. It is an interesting toping and I consider it fits to the journal. The article reports on a very interesting study that may reach large audiences. The article is very well organized, in conceptual and methodological terms, and presents very relevant results. The methodology is clearly explained. The empirical data are analysed in appropriate ways, and written up in ways that are easy to understand. The study conclusions supported are by the analysis.

Within the limits of the possibilities, the following mapping can be done:

  • Mapping of co-occurrence author keywords
  • Mapping of articles co-citations
  • Mapping of references co-citations
  • Mapping of journal co-citations
  • Mapping of institutions’ co-citations
  • Mapping of countries co-authorships
  • Mapping of countries bibliographic coupling

The bibliography can be improved. Possible sources can be:

Radu, V.; Radu, F.; Tabirca, A.I.; Saplacan, S.I.; Lile, R. Bibliometric Analysis of Fuzzy Logic Research in International Scientific Databases. Int. J. Comput. Commun. Control 2021, 16, 1–20.

Donthu, N.; Kumar, S.; Mukherjee, D.; Pandey, N.; Lim, W.M. How to conduct a bibliometric analysis: An overview and guidelines. J. Bus. Res. 2021, 133, 285–296.

Angarita-Zapata, J.S.; Maestre-Gongora, G.; Calderín, J.F. A bibliometric analysis and benchmark of machine learning and automl in crash severity prediction: The case study of three colombian cities. Sensors 2021, 21, 8401

Kaffash, S.; Nguyen, A.T.; Zhu, J. Big data algorithms and applications in intelligent transportation system: A review and bibliometric analysis. Int. J. Prod. Econ. 2021, 231, 107868

Shkundalov, D.; Vilutien, T. Bibliometric analysis of Building Information Modeling, Geographic Information Systems and Web environment integration. Autom. Constr. 2021, 128, 103757.

The authors have done an excellent job.

Reviewer 2 Report

This article is only a bibliometric analysis and has no contribution to  scholarship. It does not hace a literature review part which is a basic element in research. And it also does not have part of  conceptual framework.  In all, it is only a bibliometric analysis report from data to data.

Reviewer 3 Report

·         The novelty of this paper is not clear. The difference between present work and previous Works should be highlighted.

·         The author needs to change the abstract and focus more on problem domain. Before the paper contributions, the author could precisely include the need of developing the proposed method.

·         The author could better explain how “Related works” is actually related to the current study. It is not clear to the reader how the manuscript is similar to or differs from these related works.

·         What are all the advantages and significance of the proposed method? Mention it clearly in the proposed method.

·         The conclusion part should indicate the implications of the experimental evaluation and include some obtained values to point out the superiority clearly.

Reviewer 4 Report

The paper presents a bibliometric analysis of papers on AI and ethics published in the last 70 years. The paper is interesting and gives a detailed analysis along several axes like countries that conducted the  research, venues that published the research,  publication count and citations. It would have been more insightful if the authors analyzed the changes in research directions in this field over the years.  Although the authors have presented subject-based and keyword-based statistics, more detailed analysis especially in the light of the meteoric rise in deep learning would be useful.